# Derivatives of the β-Crinane Amaryllidaceae Alkaloid Haemanthamine as Multi-Target Directed Ligands for Alzheimer’s Disease

**DOI:** 10.3390/molecules24071307

**Published:** 2019-04-03

**Authors:** Eliška Kohelová, Rozálie Peřinová, Negar Maafi, Jan Korábečný, Daniela Hulcová, Jana Maříková, Tomáš Kučera, Loreto Martínez González, Martina Hrabinova, Katarina Vorčáková, Lucie Nováková, Angela De Simone, Radim Havelek, Lucie Cahlíková

**Affiliations:** 1ADINACO Research Group, Department of Pharmaceutical Botany, Faculty of Pharmacy, Charles University, Heyrovského 1203, 500 05 Hradec Králové, Czech Republic; kohelove@faf.cuni.cz (E.K.); perinovr@faf.cuni.cz (R.P.); negarm@faf.cuni.cz (N.M.); hulcovd@faf.cuni.cz (D.H.); 2Department of Toxicoloxy and Military Pharmacy, Faculty of Military Health Sciences, University of Defence, Třebešská 1575, 500 05 Hradec Králové, Czech Republic; korabecny.jan1@gmail.com (J.K.); tomas.kucera2@unob.cz (T.K.); hrabinova@pmfhk.cz (M.H.); 3Department Biomedical Research Centre, University Hospital Hradec Kralove, Sokolska 581, 500 05 Hradec Králové, Czech Republic; 4Department of Pharmacognosy, Faculty of Pharmacy, Charles University, Heyrovského 1203, 500 05 Hradec Králové, Czech Republic; 5Department of Organic and Bioorganic Chemistry, Faculty of Pharmacy, Charles University, Heyrovského 1203, 500 05 Hradec Králové, Czech Republic; marikoj2@faf.cuni.cz; 6Centro de Investigaciones Biológicas-CSIC, Avenida Ramiro de Maeztu 9, 28040 Madrid, Spain; loretomg@cib.csic.es; 7Deaprtment of Biological and Biochemical Sciences, Faculty of Chemical Technology, University of Pardubice, Studentská 95, 532 10 Pardubice, Czech Republic; katarina.vorcakova@upce.cz; 8Department of Analytical Chemistry, Faculty of Pharmacy, Charles University, Heyrovského 1203, 500 05 Hradec Králové, Czech Republic; novakoval@faf.cuni.cz; 9Department for Life Quality Studies, University of Bologna, Corso D’Augusto 237, 47921 Rimini, Italy; angela.desimone2@unibo.it; 10Department of Medicinal Biochemistry, Faculty of Medicine, Charles University, Zborovská 2089, 500 03 Hradec Králové, Czech Republic; havelekr@lfhk.cuni.cz

**Keywords:** haemanthamine, Amaryllidaceae, Alzheimer’s disease, acetylcholinesterase, butyrylcholinesterase, glycogen synthase kinase-3β inhibition, docking studies

## Abstract

Twelve derivatives **1a**–**1m** of the β-crinane-type alkaloid haemanthamine were developed. All the semisynthetic derivatives were studied for their inhibitory potential against both acetylcholinesterase and butyrylcholinesterase. In addition, glycogen synthase kinase 3β (GSK-3β) inhibition potency was evaluated in the active derivatives. In order to reveal the availability of the drugs to the CNS, we elucidated the potential of selected derivatives to penetrate through the blood-brain barrier (BBB). Two compounds, namely 11-*O*-(2-methylbenzoyl)-haemanthamine (**1j**) and 11-*O*-(4-nitrobenzoyl)-haemanthamine (**1m**), revealed the most intriguing profile, both being acetylcholinesterase (*h*AChE) inhibitors on a micromolar scale, with GSK-3β inhibition properties, and predicted permeation through the BBB. In vitro data were further corroborated by detailed inspection of the compounds’ plausible binding modes in the active sites of *h*AChE and *h*BuChE, which led us to provide the structural determinants responsible for the activity towards these enzymes.

## 1. Introduction

Alzheimer’s disease (AD) is the progressive neurodegenerative disease, and the one with the strongest societal impact with regard to incidence, prevalence, mortality rate, and cost of care [1]. AD is the most common cause of dementia, accounting for up 80% of all dementia cases. In 2010, about 36 million people suffered from AD worldwide, and by 2050 it is estimated that this number will reach up to 144 million AD patients [2].

AD pathogenesis is complex and comprises genetic and environmental factors that define AD as a multifactorial syndrome. Pathological changes in the brain of a patient with AD can occur more than twenty years before clinical symptoms are apparent [3]. From a neuropathological point of view, the brain of AD patients is characterized by the presence of two biomarkers: senile plaques (SPs) and neurofibrillary tangles (NFTs) of the hyperphosphorylated τ-protein [4,5]. SPs are extracellular deposits composed of various peptide fragments {Aβ_(1–40)_ and Aβ_(1–42)_} derived from the amyloid precursor protein (APP) [6]. These proteins generate deposits in specific areas of the brain, and are considered as critical factors for memory loss and cognitive impairment in AD patients [7].

Neuronal cells commonly express τ-protein, where its purpose is to stabilize the microtubules. Precise regulation of phosphorylation of τ-protein is probably important for its normocellular functioning; aberrant τ-protein hyperphosphorylation is believed to disrupt cellular processes such as axonal transport. In AD, hyperphosphorylated τ-protein accumulates and aggregates into paired helical filaments, and loses its microtubule binding and stabilizing role [8]. Phosphorylation of τ-proteins is primarily dependent on glycogen synthase kinase-3β (GSK-3β) and cyclin-dependent kinase 5 (CDK5) [9]. Genetic and epidemiological studies indicate that GSK-3β is deregulated in AD through alterations in upstream Wnt and insulin signaling pathway intermediates. [10]. GSK-3β is also cross-linked to Aβ since it may also induce its formation into aggregates [11]. However, the roles of CDK5 and GSK3 in tau hyperphosphorylation are not fully established, and it remains to determine the critical factors leading to abnormal τ-protein hyperphosphorylation.

Additionally, the disease is accompanied by cholinergic dysfunction in the CNS. The role of cholinergic neurotransmission in AD is the basis of the widely accepted cholinergic hypothesis [12,13]. Current clinical treatment of AD involves the use of reversible inhibitors of the enzyme acetylcholinesterase (AChE, E.C. 3.1.1.7) [14]. Besides posing an enhancing effect to neuronal transmission, it has also been observed that AChE accelerates the assembly of Aβ into amyloid fibrils [15]. Currently, there are three approved drugs acting as reversible inhibitors of AChE-donepezil, galanthamine and rivastigmine [14]. Similarly to AChE, another cholinesterase, butyrylcholinesterase (BuChE; E.C. 3.1.1.8) can also inactivate ACh. The later stages of AD are usually accompanied by decreased AChE activity, in contrast to BuChE level, which remains either normal or even elevated in the brain [16]. 

Traditional use of plants of the Amaryllidaceae family in folk medicine dates back to the fourth century BC, when Hippocrates of Cos used oil from the daffodil *Narcissus poeticus* L., for the treatment of uterine tumors [17]. The major secondary metabolites of the family are alkaloids, which are characterized by unique skeleton arrangements and a broad spectrum of biological activities such as antitumor [18,19], antimalarial [20], anti-inflammatory [21], antimicrobial [22], as well as AChE inhibitory activity [23]. Notably, some species of Amaryllidaceae produce the alkaloid galanthamine, which was isolated for the first time from *Galanthus woronowii* in the 1950s, and at the beginning of this century was approved for the treatment of mild to moderate stages of AD being a competitive AChE inhibitor with modulatory properties towards nicotinic ACh receptors [24]. Other alkaloids isolated from Amaryllidaceae plants like narciclasine, haemanthamine, pancratistatine and lycorine (Figure 1) have demonstrated interesting antitumor and/or apoptotic effects [18,25]. Moreover, other studies also pointed out various pharmacological properties of semisynthetic derivatives of some Amaryllidaceae alkaloids such as those related to haemanthamine [26], lycorine [27,28], narciclasine [25,29], and others (Figure 1).

Haemanthamine is a β-crinane-type alkaloid that is found in some Amaryllidaceae plants in high concentrations. Moreover, it can be easily isolated from different *Narcissus* species for semisynthetic reactions. Haemanthamine itself displays significant in vitro cytotoxic activity against several different types of cancer cell lines including MOLT-4, HepG2, HeLa, MCF7, CEM, K562, A549 [30], Caco-2, HT-29 [31], A278, SW1573 and T47-D [25].

Nowadays the portfolio of drugs used for treatment of AD is narrow [14], and the search for new and effective therapy of the disease is urgently needed. As a part of our ongoing research on Amaryllidaceae alkaloids being potential drugs in the treatment of neurodegenerative diseases, we have carried out modifications of the structure of the β-crinane type alkaloid haemanthamine (**1**). The obtained derivatives were evaluated for their potency for the inhibition of enzymes connected with the potential treatment of AD (*h*AChE, *h*BuChE, and GSK-3β). Drugs targeting the CNS need to be able to pass through the blood-brain barrier (BBB), and thus the potential of the selected derivatives to penetrate this was also studied. In vitro data were further corroborated by detailed inspection of the drugs’ plausible binding modes in the active sites of *h*AChE and *h*BuChE, which allowed us to establish some preliminary structure-activity relationships.

## 2. Results

### 2.1. Synthesis of Haemanthamine Derivatives ***1a**–**1m***

The structural aspects of haemanthamine (**1**) allowed us to prepare novel chemical entities by derivatization of its free hydroxyl group and to inspect the structure-activity relationship of the novel derivatives. Structural modifications of the haemanthamine scaffold have not been previously described in the literature. Only a few marginal studies exist, one of them reporting alkaline degradation of oxohaemanthamine and oxodihydrohaemanthamine to afford haemanthamine [32]. The scarcity of the literature further underlies a report dealing with stereoselective tyramine labeling, which was used to detect enzymatic hydroxylation of haemanthamine [33]. Noteworthy, some aliphatic esters of haemanthamine or dihydrohaemanthamine have recently been proposed to alter fermentation in the rumen [34]. The chemical modifications of **1** were possible because of its large scale isolation from *Narcissus pseudonarcissus* cv. Dutch Master. The preparation of **1** derivatives was inspired by lycorine, another biologically active Amaryllidaceae alkaloid. Lycorine itself exerted very low inhibitory activity against cholinesterases, but its acylated and etherified analogues possessed potent inhibitory activities against both cholinesterases [35,36]. In this context, aliphatic and aromatic substituents (esters) were explored in order to establish a structure-activity relationship (SAR). All the structural modifications related to **1** are shown in Scheme 1. The substitutions were carried out on the hydroxyl group at C-11. Compounds **1a**–**1e** were obtained by acylation with corresponding anhydrides (Scheme 1) [26]. The hydroxyl group was also acylated with differently substituted benzoyl chlorides affording the corresponding esters **1f**–**1m** (Scheme 1). The yields of the reactions exceeded 65 % in all cases.

### 2.2. Biological Profile of Haemanthamine and Haemanthamine Derivatives ***1a**–**1m***

The primary goal of the development of **1** derivatives was to investigate them as novel chemical probes with potential higher cytotoxic activity against tumor cell lines and compare them with the parent compound **1**, while maintaining low toxicity to healthy cells. As reported previously, **1** displayed pronounced cell growth inhibitory activities against a variety of tumor cells [25,30,31]. This action is presumably mediated by inhibiting early stages of ribosome biogenesis via p53-dependent antitumoral stress response [37]. Up to date, there are only a few studies devoted to drug design and biological activities of the derivatives of **1 [25,26]**. Interestingly, **1** was found to be completely inactive to both cholinesterases. In the first step we decided to prepare a library of six aliphatic (compounds **1a**–**1e**) and six aromatic esters (compounds **1g**–**1m**). Out of these compounds, only two (**1a**, **1c**) have been described previously [26], while the others have been synthesized for the first time in the current study. The antiproliferative effect of **1a**–**1m** at 10 µM single-dose treatment regimen was assessed using WST-1 tetrazolium salt assay. Interestingly, none of the derivatives displayed any considerable toxicity against the tested human cell lines (Jurkat, MOLT-4, A549, HT-29, PANC-1, A2780, HeLa, MCF-7, SAOS-2 and MRC-5) during 48 h of exposure, which is in striking contrast with **1** that elicited potent cytotoxic and apoptotic effects. Mean values for each cell line proliferation are shown in Appendix A. In the next step, the low cytotoxicity of the prepared derivatives led us to investigate the interaction of **1a**–**1m** with cholinesterases, and GSK-3β. Interestingly, some derivatives exerted promising inhibitory activities against cholinesterases (Table 1) and GSK-3β. All aliphatic esters **1a**–**1f** displayed either weak (1 mM ≥ IC_50_ ≥ 200 µM) or no (IC_50_ ≥ 1000 µM) inhibition activity in the cholinesterase assay. The presence of an aromatic acyl-, as in compounds **1g**–**1m**, seems to be crucial for the *h*AChE and *h*BuChE inhibitory potency. The most active inhibitors of *h*AChE were 11-*O*-(4-nitrobenzoyl)haemanthamine (**1m**) and 11-*O*-(2-methylbenzoyl)haemanthamine (**1j**), both displaying two-digit micromolar IC_50_ values (Table 1). The unsubstituted analogue 11-*O*-benzoylhaemanthamine (**1g**) revealed reduced potency by one order of magnitude. Analogue 11-*O*-(2-methylbenzoyl)haemanthamine (**1j**) showed a non-selective profile for both cholinesterases in the micromolar scale. Kinetic analysis of **1j** and **1m** was used for the determination of their inhibition pattern against both *h*AChE and *h*BuChE. Results are shown using Lineweaver-Burk plots (Figure 2) for **1j** [38]. For the results of kinetic analysis for **1m**, readers are kindly referred to the Appendix A. The *K*_m_ and *V*_m_ values were calculated from the Lineweaver-Burk plot. The values of K_m_ and V_m_ for reactions in the presence of the tested alkaloids were decreased compared with the values for the reaction in their absence. From the results obtained it could be concluded that both derivatives act via a mixed mechanism of inhibition. Corresponding *K*_i_ values of 60.1 ± 0.68 μM and 12.5 ± 0.45 μM were determined for *h*AChE and *h*BuChE, respectively.

Selected haemanthamine derivatives (compounds **1g**, **1j** and **1m**) were screened for their GSK-3β inhibition potency at a concentration of 10 µM. Since **1m** revealed the highest inhibition potency in the tested concentration scale, its IC_50_ was further determined demonstrating a slightly less potent profile compared with the native Amaryllidaceae alkaloids masonine, caranine, and narcimatuline [39].

The crucial part in the process of AD therapeutic drug development is the ability of the drug to reach the CNS, thus crossing the BBB. The screening for BBB penetration in early drug discovery programs provides important information for compound selection [40]. The majority of CNS drugs enter the brain by transcellular passive diffusion, thus, the evaluation of ADME (absorption, distribution, metabolism, excretion) properties is of importance to reduce attrition in the development process. The parallel artificial membrane permeability assay (PAMPA) was selected as a useful technique to predict passive diffusion through biological membranes [41], employing a brain lipid porcine membrane. The in vitro permeability (*P*e) through the lipid membrane of commercial drugs was determined together with compounds **1g**, **1j** and **1m** (Table 1). An assay validation was made comparing the reported permeability values of the commercial drugs with our experimental data (see Appendix A). A good correlation between experimental and described values was obtained *P*e (exptl) = 0.9079 (bibl) − 0.696 (R^2^ = 0.974). Following the pattern established in the literature for BBB permeation prediction we could classify compounds under the survey (**1g**, **1j** and **1m**) as centrally active having *P*e values > 4.00 × 10^−6^ cm s^−1^ (Table 1). However, the PAMPA assay uses artificial membranes to observe passive membrane permeability, thus neglecting the special characteristics of the BBB. For this reason we also exploited another methodology calculating logBB for active haemanthamine derivatives **1g**, **1j** and **1m**. LogBB is the most common numeric value describing permeability across the BBB. It is defined as the logarithmic ratio between the concentration of a compound in the brain (C*_brain_*) and blood (C*_blood_*) [42]. LogBB values of the calculated compounds vary from 0.007 to 0.285. Compounds with logBB > 0.3 readily cross the BBB, while those with logBB < −1.0 are only poorly distributed to the brain [43]. The obtained results for both the PAMPA assay and logBB calculation indicated that the novel derivatives represent compounds with potential to permeate through the BBB.

### 2.3. Docking Studies

In order to understand better the structural basis of *h*AChE and *h*BuChE inhibition by the new semi-synthetic alkaloids reported herein, we have conducted a computational study to identify a binding mode allowing us to account for the experimental SAR. All the compounds were firstly submitted to a screening method employing docking with flexible residues with 20 repetitions for each ligand **1a**–**1m** counting both cholinesterases, i.e., *h*AChE (PDB ID: 4EY7) and *h*BuChE (PDB ID: 4BDS) [44,45]. The selection of these enzymes is in accordance with the in vitro testing where enzymes of human origin were also used. The binding energies for the top-scored docking poses are collected in Table 2. The results for *h*AChE nicely fit those obtained in vitro, even though this has to be taken with precaution since no clear correlation has been established between the scoring functions from in silico and data from in vitro [46]. In the case of *h*AChE inhibition, more potent inhibitors, i.e., those possessing IC_50_s in the two to three digit micromolar range (compounds **1g**–**1m**), displayed binding energies between −12.7 kcal/mol to −13.4 kcal/mol. This stems from additional hydrophobic-type interaction, mainly mediated by π-π interactions with Tyr337 or Trp86, given by the presence of an attached aryl moiety to the basic core of **1**. A similar trend can be found when dealing with in silico results for *h*BuChE (Table 2). It is apparent that potent enzyme inhibition is linked to the presence of an aryl appendage attached to **1**, which is also reflected by the calculations for **1g**–**1k** (binding energy estimates ≥ −12.8 kcal/mol). With the exception of **1m**, this is due to aryl-aryl interactions mediated between the arylcarbonyl moiety of the ligands and Phe329 (this type of interaction is missing in the case of aliphatic derivatives **1a**–**1f**) and by other less-specific interactions, mostly with catalytic triad residues.

Next, we turned our attention to two of the most pronounced cholinesterase inhibitors, namely **1j** and **1m**. The rationale for their selection lies in the fact that **1j** was the most active *h*BuChE and non-specific inhibitor, whereas **1m** exerted the highest affinity and selectivity for *h*AChE. In accordance with in vitro testing, these ligands were docked into enzyme cavities of human origin. All 20 poses retrieved from calculation via Autodock Vina software mostly preserved the putative binding modes coined to the catalytic anionic site (CAS) region of both cholinesterases (Table 2). This applies to both **1j** (−13.8 kcal/mol for *h*AChE and −13.7 kcal/mol for *h*BuChE) and **1m** (−13.4 kcal/mol for *h*AChE and −11.7 kcal/mol for *h*BuChE).

As expected, **1j** (Figure 3A,B) is anchored within the CAS region at the bottom of the cavity gorge of *h*AChE. The 2-methylbenzoyl moiety is sandwiched between Tyr337 (parallel π-π stacking, 3.7 Å) and Tyr341 (distorted π-π stacking, 3.7 Å). 

Some π-alkyl interaction can also be observed between Phe338 and the 2-methylbenzoyl moiety. Linking the carbonyl from the ester group gave rise to a hydrogen bond with the hydroxyl from Tyr124 (3.1 Å). The core of the ligand is located in the proximity of aromatic residues Trp86 and Tyr133 via van der Waal’s interaction. More importantly, His447 enabled the formation of a cation-π bond with a protonated nitrogen from the tetrahydroisoquinoline scaffold of **1j** (3.7 Å). Another residue from the catalytic triad, namely Ser203, displayed van der Waal’s contact with the polycyclic aliphatic cage of **1j**. Compound **1m** (Figure 3C,D) revealed a different binding pose in the *h*AChE active site compared with **1j**, with the heamanthamine core of **1m** protruding the out of the gorge towards the peripheral anionic site (PAS). This structural alteration is plausibly the result of several interactions responsible for the ligand docking, namely i) hydrogen bond formation between the 4-nitro group of the benzoyl moiety and the hydroxyl from Tyr133 (3.0 Å and 2.9 Å), ii) π-π stacking with Tyr341 (3.9 Å) and Phe297 (3.8 Å) in a parallel and T-shaped manner, respectively, iii) CH-π and cation-π forces with Phe338 and iv) parallel π-π interaction between the 4-nitrobenzoyl moiety and Trp86 (3.7 Å). The catalytic triad in this case remained unaffected; only Ser203 seems to be employed in some non-specific hydrophobic interactions with **1m**. In general, it can be concluded that a major discrepancy in the binding pose is given by the presence of the nitro group in the 4-nitrobenzoyl moiety. It is noteworthy that both **1j** and **1m** arranged in completely different binding poses compared with galantamine in the *h*AChE active site (PDB ID: 4EY6), but still fitted well in the CAS region [44]. This finding could be a promising starting point in terms of designing novel ligands derived from haemanthamine.

Next, we inspected the determinants responsible for the binding differences between **1j**-*h*BuChE (Figure 4A,B) and **1m**-*h*BuChE (Figure 4C,D) complexes. Similar binding patterns for both ligands can be observed in the *h*BuChE active site, i.e., i) the tetrahydroisoquinoline moiety imposed face-to-face π-π stacking to Trp82 at distances of 3.7 Å and 4.3 Å for **1j** and **1m**, respectively, ii) a salt bridge between Asp70 (2.2 Å and 2.6 Å for **1j** and **1m**, respectively) and a protonated nitrogen of the haemanthamine core, as well as iii) a benzoyl moiety residing in the proximity of Phe329 via T-shaped π-π contacts (3.7 Å for both ligands). The possible explanation of the different affinities of each ligand may be the engagement of the catalytic triad residues in the ligand-enzyme lodging, as well as the overall hydrophobic contribution. In this regard, all three catalytic triad residues, i.e., Glu197, Ser198 and His438, are affected by **1j**, whereas contact with Glu197 is missing in the case of the **1m**-*h*BuChE complex. On the other hand, the nitro group of **1m** provided a conventional hydrogen bond with the hydroxyl from Ser198 (2.9 Å). As indicated above, other hydrophobic contributors like π-alkyl interaction with Tyr332 presumably provided better fitting in the *h*BuChE active site. In this respect, **1j** can be considered a more balanced, non-selective cholinesterase inhibitor worthy of further studies.

## 3. Experimental

### 3.1. General Experimental Procedures

All solvents were treated by using standard techniques before use. All reagents and catalysts were purchased from commercial sources (Sigma Aldrich, Prague, Czech Republic) and used without purification. The NMR spectra were obtained in CDCl_3_ and CD_3_OD at ambient temperature on a VNMR S500 (Varian, Palo Alto, CA, USA) spectrometer operating at 500 MHz for ^1^H and 125.7 MHz for ^13^C. Chemical shifts were recorded as *δ* values in parts per million (ppm) and were indirectly referenced to tetramethylsilane (TMS) via the solvent signal (CDCl_3_–7.26 ppm for ^1^H and 77.0 ppm for ^13^C; CD_3_OD–3.30 ppm for ^1^H and 49.0 ppm for ^13^C). Coupling constants (*J*) are given in Hz. For unambiguous assignment of ^1^H and ^13^C signals, 2D NMR experiments, namely gCOSY, gHSQC, gHMBC and NOESY were measured using standard parameter settings and standard pulse programs provided by the producer of the spectrometer. ESI-HRMS spectra were obtained with a Synapt G2-Si hybrid mass analyzer of a quadrupole-time-of-flight (Q-TOF) type (Waters, Manchester, Great Britain) coupled to a Waters Acquity I-Class UHPLC system. The EI-MS were obtained on an Agilent 7890A GC 5975 inert MSD operating in EI mode at 70 eV (Agilent Technologies, Santa Clara, CA, USA). A DB-5 column (30 m × 0.25 mm × 0.25 μm, Agilent Technologies) was used. The temperature program was: 100–180 °C at 15 °C/min, 1 min hold at 180 °C, and 180–300 °C at 5 °C/min and 5 min hold at 300 °C; detection range *m*/*z* 40–600. The injector temperature was 280 °C. The flow-rate of the carrier gas (helium) was 0.8 mL/min. A split ratio of 1:15 was used. TLC was carried out on precoated silica gel 60 F254 plates (Merck, Darmstadt, Germany). Compounds on the plate were observed under UV light (254 and 366 nm) and visualized by spraying with Dragendorff’s reagent. 

### 3.2. Preparation of Haemanthamine Derivatives

Haemanthamine (**1**) used for preparation of semisynthetic derivatives has been isolated from fresh bulbs of *Narcissus pseudonarcissus* cv. Dutch Master (unpublished results, for ^1^H-, ^13^C-NMR, HPLC, and HPLC-MS spectra see Appendix A).

#### 3.2.1. General Procedure for Acylation of 1 Using Anhydrides

To a solution of haemanthamine (**1**) in 3 mL of dry pyridine, 2.5–5 eq. of the corresponding anhydride was added. The mixture was stirred at room temperature until the starting material disappeared (TLC detection). The solvent was then evaporated and the residue was purified by preparative TLC using either cHx:Et_2_NH 9:1 or cHx:To:Et_2_NH 45:45:10 to afford corresponding esters **1a**–**1e.**

##### 11-*O*-Acetylhaemanthamine (**1a**)

To **1** (50 mg, 0.164 mmol) in pyridine (3 mL), acetic anhydride (800 µL, 0.846 mmol) was added. After 3 h stirring, the solvent was evaporated and the residue was purified by preparative TLC using To:Et_2_NH 9:1 to yield 54 mg (92%) of **1a** as an amorphous white solid. The NMR and MS data were in accordance with published data [23].

##### 11-*O*-Propionylhaemanthamine (**1b**)

To **1** (50 mg, 0.164 mmol) in pyridine (3 mL), propionic anhydride (100 µL, 0.779 mmol) was added. After 6 h stirring, the solvent was evaporated and the residue was purified by preparative TLC using To:Et_2_NH 9:1 to yield 51 mg (87 %) of **1b** as colorless oil. ^1^H-NMR (500 MHz, CDCl_3_) δ: 6.90 (1H, s), 6.46 (1H, s), 6.34 (1H, d, *J* = 10.3 Hz), 6.14 (1H, dd, *J* = 10.3 Hz, *J* = 4.9 Hz), 5.90–5.88 (2H, m), 4.97 (1H, dd, *J* = 6.8 Hz, *J* = 3.4 Hz), 4.35 (1H, d, *J* = 17.1 Hz), 3.85–3.81 (1H, m), 3.71 (1H, d, *J* = 17.1 Hz), 3.42–3.28 (2H, m), 3.35 (3H, s), 2.31–2.18 (2H, m), 2.13–2.01 (2H, m), 1.94 (1H, td, *J* = 13.7Hz, *J* = 4.4Hz), 1.09 (3H, t, *J* = 7.8Hz). ^13^C-NMR (125 MHz, CDCl_3_) δ: 173.3, 146.6, 146.4, 134.3, 129.4, 127.7, 126.5, 106.5, 103.9, 100.8, 80.0, 72.6, 62.8, 61.2, 60.6, 56.5, 49.2, 28.3, 27.8, 9.1. ${\left[\alpha \right]}_{D}^{24}$ = +10.8 (*c* = 0.230; CHCl_3_). EI-MS *m*/*z* (%) 357 (100), 283 (26), 268 (29), 252 (26), 240 (22), 224 (63), 210 (34), 181 (38). ESI-HRMS *m*/*z* calcd for C_20_H_23_NO_5_ [M + H]^+^ 358.1654 found 358.1656.

##### 11-*O*-Isobutanoylhaemanthamine (**1c**)

To **1** (50 mg, 0.164 mmol) in pyridine (3 mL), isobutyric anhydride (100 µL, 0.603 mmol) was added. After 8 h stirring, the solvent was evaporated and the residue was purified by preparative TLC using To:cHx:Et_2_NH 45:45:10 to yield 52 mg (85 %) of **1c** as an amorphous white solid. The NMR and MS data were in accordance with published data [23].

##### 11-*O*-Pentanoylhaemanthamine (**1d**)

To **1** (50 mg, 0.164 mmol) in pyridine (3 mL), valeric anhydride (100 µL, 0.506 mmol) was added. After 8 h stirring, the solvent was evaporated and the residue was purified by preparative TLC using To:cHx:Et_2_NH 30:70:5 as eluent to yield 54 mg (85 %) of **1d** as a pale yellow oil. ^1^H-NMR (500 MHz, CDCl_3_) δ: 6.91 (1H, s), 6.47 (1H, s), 6.35 (1H, d, *J* = 10.3 Hz), 6.15 (1H, dd, *J* = 10.3 Hz, *J* = 4.9 Hz), 5.91-5.88 (2H, m), 4.98 (1H, dd, *J* = 7.3 Hz, *J* = 2.9 Hz), 4.36 (1H, d, *J* = 16.6 Hz), 3.86–3.81 (1H, m), 3.73 (1H, d, *J* = 16.6 Hz), 3.43–3.29 (3H, m), 3.36 (3H, s), 2.22 (2H, td, *J* = 7.3 Hz, *J* = 3.9 Hz), 2.08–2.03 (1H, m), 1.98–1.91(1H, m), 1.59-1.50 (2H, m), 1.38–1.28 (2H, m), 0.91 (3H, t, *J* = 7.3 Hz). ^13^C-NMR (125 MHz, CDCl_3_) δ: 172.7, 146.7, 146.4, 134.3, 129.5, 127.7, 126.4, 106.6, 103.9, 100.9, 80.0, 72.6, 62.8, 61.1, 60.6, 56.5, 49.2, 34.2, 28.3, 26.9, 22.2, 13.7. ${\left[\alpha \right]}_{D}^{24}$ = +15.7 (*c* = 0.508; CHCl_3_). EI-MS *m*/*z* (%) 385 (100), 356 (10), 300 (12), 284 (28), 283 (28), 268 (45), 252 (30), 240 (29), 224 (70), 210 (48), 181 (46). ESI-HRMS *m*/*z* calcd for C_22_H_27_NO_5_ [M + H]^+^ 386.1967 found 386.1968.

##### 11-*O*-Hexanoylhaemanthamine (**1e**)

To **1** (50 mg, 0.164 mmol) in pyridine (3 mL), hexanoic anhydride (100 µL, 0.433 mmol) was added. After 12 h stirring, the solvent was evaporated and the residue was purified by preparative TLC using To:cHx:Et_2_NH 30:70:5 as eluent to yield 59 mg (90 %) of **1e** as a colorless oil. ^1^H-NMR (500 MHz, CDCl_3_) δ: 6.91 (1H, s), 6.46 (1H, s), 6.35 (1H, d, *J* = 10.0 Hz), 6.14 (1H, ddd, *J* = 10.0 Hz, *J* = 4.7 Hz, *J* = 1.0 Hz), 5.90–5.88 (2H, m), 4.97 (1H, ddd, *J* = 7.4 Hz, *J* = 3.4 Hz, *J* = 1.0 Hz), 4.35 (1H, d, *J* = 16.8 Hz), 3.85–3.81 (1H, m), 3.71 (1H, d, *J* = 16.8 Hz), 3.44–3.26 (3H, m), 3.36 (3H, s), 2.24-2.18 (2H, m), 2.07–2.01 (1H, m), 1.94 (1H, td, *J* = 13.7 Hz, *J* = 4.7 Hz), 1.60–1.53 (2H, m), 1.36–1.23 (4H, m), 0.90 (3H, t, *J* = 6.8 Hz). ^13^C-NMR (125 MHz, CDCl_3_) δ: 172.7, 146.6, 146.4, 134.4, 129.4, 127.8, 126.6, 106.5, 103.9, 100.8, 80.1, 72.6, 62.8, 61.2, 60.7, 56.5, 49.2, 34.4, 31.2, 28.4, 24.5, 22.2, 13.8. ${\left[\alpha \right]}_{D}^{24}$ = +9.8 (*c* = 0.449; CHCl_3_). EI-MS *m*/*z* (%) 399 (100), 370 (10), 300 (10), 284 (37), 269 (51), 252 (34), 240 (35), 224 (74), 209 (57), 181 (50). ESI-HRMS *m*/*z* calcd for C_23_H_29_NO_5_ [M + H]^+^ 400.2124 found 400.2126.

#### 3.2.2. General Procedure for Acylation of **1** Using Chlorides

To a solution of **1** (50 mg, 0.164 mmol) in dry pyridine (3 mL), 1.5–3.0 eq. of the corresponding acyl chloride and a catalytic amount of 4-dimethylaminopyridine (DMAP; 2 mg) were added to the reaction. The reaction mixture was stirred at 80 °C until disappearance of the starting material (5–20 h). The solvent was then evaporated and the residue was purified by preparative TLC using cHx:To:Et_2_NH 45:45:10 to afford the corresponding esters **1f**–**1m.**

##### 11-*O*-Butanoylhaemanthamine (**1f**)

The procedure described above was followed using butyryl chloride (50 µL, 0.480 mmol) and a catalytic amount of DMAP (2 mg). After 12 h stirring at 80 °C, the solvent was evaporated and the residue was purified by preparative TLC to yield 54 mg (89%) of **1f** as a colorless oil. ^1^H-NMR (500 MHz, CDCl_3_) δ: 6.90 (1H, s), 6.45 (1H, s), 6.34 (1H, d, *J* = 9.8 Hz), 6.13 (1H, dd, *J* = 9.8 Hz, *J* = 4.9 Hz), 5.91–5.88 (2H, m), 4.96 (1H, dd, *J* = 6.9 Hz, *J* = 3.4 Hz), 4.34 (1H, d, *J* = 17.1 Hz), 3.85-3.81 (1H, m), 3.70 (1H, d, *J* = 17.1 Hz), 3.42-3.26 (3H, m), 3.35 (3H, s), 2.19 (2H, td, *J* = 7.8 Hz, *J* = 4.4 Hz), 2.05–2.00 (1H, m), 1.93 (1H, td, *J* = 13.7 Hz, *J* = 4.4 Hz), 1.63–1.54 (2H, m), 0.93 (3H, t, *J* = 7.3 Hz).^13^C-NMR (125 MHz, CDCl_3_) δ: 172.6, 146.6, 146.4, 134.4, 129.4, 127.8, 126.6, 106.6, 103.9, 100.8, 80.1, 72.6, 62.8, 61.2, 60.7, 56.5, 49.2, 36.4, 28.4, 18.3, 13.7. ${\left[\alpha \right]}_{D}^{24}$ = +25.6 (*c* = 0.110; CHCl_3_). EI-MS *m*/*z* (%) 371 (100), 284 (30), 268 (40), 252 (38), 240 (37), 224 (73), 210 (46), 181 (47), 153 (21), 115 (23). ESI-HRMS *m*/*z* calcd for C_21_H_25_NO_5_ [M + H]^+^ 372.1811 found 372.1813.

##### 11-*O*-Benzoylhaemanthamine (**1g**)

The procedure described above was followed using benzoyl chloride (50 µL, 0.430 mmol) and a catalytic amount of DMAP (2 mg). After 8 h stirring at 80 °C, the solvent was evaporated and the residue was purified by preparative TLC using To:cHx:Et_2_NH 60:45:5 as eluent to yield 56 mg (85 %) of **1g** as an amorphous white solid. ^1^H-NMR (500 MHz, CDCl_3_) δ: 8.12–8.07 (1H, m), 7.94–7.90 (2H, m), 7.59–7.54 (1H, m), 7.46–7.41 (2H, m), 6.96 (1H, s), 6.51 (1H, s), 6.43 (1H, d, *J* = 10.0 Hz), 6.13 (1H, dd, *J* = 10.0 Hz, *J* = 4.9 Hz), 5.91 (1H, d, overlapped, *J* = 4.4 Hz), 5.91 (1H, d, overlapped, *J* = 4.4 Hz), 4.44 (1H, d, *J* = 16.9 Hz), 3.89–3.85 (1H, m), 3.82 (1H, d, *J* = 16.9 Hz), 3.57–3.54 (2H, m), 3.49 (1H, dd, *J* = 13.2 Hz, *J* = 4.4 Hz), 3.35 (3H, s), 2.23–2.18 (1H, m), 2.13 (1H, td, *J* = 13.2 Hz, *J* = 4.4 Hz). ^13^C-NMR (125 MHz, CDCl_3_) δ: 165.4, 146.8, 146.6, 134.0, 133.1, 131.9, 130.0, 129.6, 129.3, 128.4, 128.0, 127.4, 126.0, 106.7, 103.9, 100.9, 80.6, 72.3, 62.8, 60.9, 60.5, 56.5, 49.3, 28.2. ${\left[\alpha \right]}_{D}^{24}$ = +42.1 (*c* = 0.133; CHCl_3_). EI-MS *m*/*z* (%) 405 (55), 300 (6), 283 (38), 268 (33), 252 (22), 224 (62), 210 (29), 181 (24), 105 (100), 77 (53). ESI-HRMS *m*/*z* calcd for C_24_H_23_NO_6_ [M + H]^+^ 406.1654 found 406.1658.

##### 11-*O*-(3-Chlorobenzoyl)haemanthamine (**1h**)

The procedure described above was followed using 3-chlorobenzoyl chloride (50 µL, 0.391 mmol) and a catalytic amount of DMAP (2 mg). After 8 h stirring at 80 °C, the solvent was evaporated and the residue was purified by preparative TLC to yield 56 mg (85%) of **1h** as an amorphous white solid. ^1^H-NMR (500 MHz, CDCl_3_) δ: 7.88 (1H, t, *J* = 1.6 Hz), 7.80 (1H, dt, *J* = 7.8 Hz, *J* = 1.6 Hz), 7.55–7.52 (1H, m), 7.38 (1H, t, *J* = 7.8 Hz), 6.95 (1H, s), 6.50 (1H, s), 6.42 (1H, d, *J* = 10.0 Hz), 6.13 (1H, dd, *J* = 10.0 Hz, *J* = 4.9 Hz), 5.91 (1H, d, overlapped, *J* = 4.4 Hz), 5.91 (1H, d, overlapped, *J* = 4.4Hz), 5.19 (1H, dd, *J* = 7.0 Hz, *J* = 3.4 Hz), 4.39 (1H, d, *J* = 17.1Hz), 3.89–3.85 (1H, m), 3.77 (1H, d, *J* = 17.1 Hz), 3.54 (1H, dd, *J* = 14.2 Hz, *J* = 7.2 Hz), 3.48–3.41 (2H, m), 3.37 (3H, s), 2.17–2.11 (1H, m), 2.04 (1H, td, *J* = 13.7 Hz, *J* = 4.2 Hz). ^13^C-NMR (125 MHz, CDCl_3_) δ: 164.3, 146.7, 146.5, 134.6, 134.1, 133.1, 131.9, 129.8, 129.7, 129.4, 127.6, 127.4, 126.6, 106.6, 103.9, 100.9, 81.3, 72.4, 62.9, 61.2, 60.9, 56.5, 49.2, 28.6. ${\left[\alpha \right]}_{D}^{24}$ = +45.5 (*c* = 0.167; CHCl_3_). EI-MS *m*/*z* (%) 439 (65), 300 (8), 283 (39), 270 (35), 252 (40), 224 (88), 181 (31), 141 (40), 139 (100), 111 (48), 75 (15). ESI-HRMS *m*/*z* calcd for C_24_H_22_ClNO_5_ [M + H]^+^ 440.1265 found 440.1272.

##### 11-*O*-(3-Bromobenzoyl)haemanthamine (**1i**)

The procedure described above was followed using 3-bromobenzoyl chloride (50 µL, 0.379 mmol) and a catalytic amount of DMAP (2 mg). After 8 h stirring at 80 °C, the solvent was evaporated and the residue was purified by preparative TLC to yield 75 mg (93%) of **1i** as a pale white oil. ^1^H- NMR (500 MHz, CDCl_3_) δ: 8.04–8.01 (1H, m), 7.84–7.81 (1H, m), 7.68–7.65 (1H, m), 7.30 (1H, t, *J* = 7.8 Hz), 6.93 (1H, s), 6.48 (1H, s), 6.40 (1H, d, *J* = 9.8 Hz), 6.12 (1H, dd, *J* = 9.8 Hz, *J* = 4.9 Hz), 5.90–5.88 (2H, m), 5.17 (1H, dd, *J* = 6.8 Hz, *J* = 3.4 Hz), 4.38 (1H, d, *J* = 16.6 Hz), 3.87–3.83 (1H, m), 3.75 (1H, d, *J* = 16.6 Hz), 3.55–3.48 (1H, m), 3.46–3.40 (2H, m), 3.35 (3H, s), 2.13 (1H, dd, *J* = 13.5 Hz, *J* = 4.4 Hz), 2.02 (1H, td, *J* = 13.5 Hz, *J* = 4.4 Hz). ^13^C-NMR (125 MHz, CDCl_3_) δ: 164.0, 146.6, 146.5, 135.9, 134.0, 132.3, 132.0, 129.9, 129.6, 127.7, 127.5, 126.5, 122.4, 106.6, 103.8, 100.8, 81.2, 72.3, 62.8, 61.1, 60.8, 56.5, 49.2, 28.5. ${\left[\alpha \right]}_{D}^{24}$ = +66.9 (*c* = 0.202; CHCl_3_). EI-MS *m*/*z* (%) 485 (46), 483 (44), 300 (10), 283 (50), 268 (43), 252 (46), 240 (21), 238 (20), 224 (100), 210 (45), 184 (75), 182 (82), 154 (40), 115 (20), 76 (25). ESI-HRMS *m*/*z* calcd for C_24_H_22_BrNO_5_ [M + H]^+^ 484.0760 found 484.0763.

##### 11-*O*-(2-Methylbenzoyl)haemanthamine (**1j**)

The procedure described above was followed using 2-methylbenzoyl chloride (50 µL, 0.383 mmol) and a catalytic amount of DMAP (2 mg). After 8 h stirring at 80 °C, the solvent was evaporated and the residue was purified by preparative TLC to yield 45 mg (65%) of **1j** as an amorphous white solid. ^1^H-NMR (500 MHz, CDCl_3_) δ: 7.76–7.73 (1H, m), 7.42–7.38 (1H, m), 7.25–7.21 (2H, m), 6.98 (1H, s), 6.50 (1H, s), 6.46 (1H, d, *J* = 10.0 Hz), 6.15 (1H, dd, *J* = 10.0 Hz, *J* = 4.9 Hz), 5.91 (2H, s), 5.20 (1H, dd, *J* = 6.8 Hz, *J* = 3.9 Hz), 4.41 (1H, d, *J* = 16.6 Hz), 3.87–3.84 (1H, m), 3.78 (1H, d, *J* = 16.6 Hz), 3.56–3.43 (3H, m), 3.36 (3H, s), 2.57 (3H, s), 2.17–2.11 (1H, m), 2.10–2.03 (1H, m). ^13^C-NMR (125 MHz, CDCl_3_) δ: 166.3, 146.7, 146.5, 140.3, 134.3, 132.0, 131.7, 130.0, 129.5, 129.3, 127.8, 126.3, 125.7, 106.6, 103.9, 100.9, 80.7, 72.4, 62.8, 61.0, 60.7, 56.5, 49.3, 28.3, 21.6. ${\left[\alpha \right]}_{D}^{24}$ = +33.0 (*c* = 0.135; CHCl_3_). EI-MS *m*/*z* (%) 419 (30), 283 (17), 268 (28), 225 (35), 210 (15), 119 (100), 91 (40). ESI-HRMS *m/z* calcd for C_25_H_25_NO_5_ [M + H]^+^ 420.1811 found 420.1812.

##### 11-*O*-(3-Methoxybenzoyl)haemanthamine (**1k**)

The procedure described above was followed using 3-methoxylbenzoyl chloride (50 µL, 0.344 mmol) and a catalytic amount of DMAP (2 mg). After 8 h stirring at 80 °C, the solvent was evaporated and the residue was purified by preparative TLC using To:Et_2_NH 9:1 as eluent to yield 69 mg (95%) of **1k** as a pale white oil. ^1^H-NMR (500 MHz, CDCl_3_) δ: 7.49 (1H, d, *J* = 7.9 Hz), 7.46–7.42 (1H, m), 7.32 (1H, t, *J* = 7.9 Hz), 7.09 (1H, dd, *J* = 7.9 Hz, *J* = 2.4 Hz), 6.94 (1H, s), 6.48 (1H, s), 6.42 (1H, d, *J* = 10.0 Hz), 6.12 (1H, dd, *J* = 10.0 Hz, *J* = 4.9 Hz), 5.90–5.87 (2H, m), 5.18 (1H, dd, *J* = 10.0 Hz, *J* = 3.4 Hz), 4.38 (1H, d, *J* = 16.6 Hz), 3.87–3.84 (1H, m), 3.82 (3H, s), 3.80 (1H, d, *J* = 16.6 Hz), 3.55–3.39 (3H, m), 3.34 (3H, s), 2.15–2.00 (2H, m). ^13^C-NMR (125 MHz, CDCl_3_) δ: 165.2, 159.5, 146.6, 146.4, 134.2, 131.3, 129.5, 129.4, 127.6, 126.5, 121.5, 119.3, 114.0, 106.6, 103.8, 100.8, 80.9, 72.4, 62.8, 61.1, 60.9, 56.4, 55.3, 49.1, 28.5. ${\left[\alpha \right]}_{D}^{24}$ = +36.3 (*c* = 0.265; CHCl_3_). EI-MS *m*/*z* (%) 435 (45), 300 (7), 283 (38), 268 (35), 252 (20), 224 (55), 210 (26), 181 (17), 152 (14), 135 (100), 107 (28), 92 (15), 77 (25.) ESI-HRMS *m*/*z* calcd for C_25_H_25_NO_6_ [M + H]^+^ 436.1760 found 436.1761.

##### 11-*O*-(4-Nitrobenzoyl)haemanthamine (**1m**)

The procedure described above was followed using 4-nitrobenzoyl chloride (46 mg, 0.246 mmol) and a catalytic amount of DMAP (2 mg). After 8 h stirring at 80 °C, the solvent was evaporated and the residue was purified by preparative TLC using To:cHx:Et_2_NH 45:45:10 as eluent to yield 48 mg (65%) of **1m** as an amorphous yellow solid. ^1^H-NMR (500 MHz, CDCl_3_) δ: 8.31–8.28 (2H, m, AA´BB´), 8.11–8.07 (2H, m, AA´BB´), 6.95 (1H, s), 6.51 (1H, s), 6.42 (1H, d, *J* = 10.3 Hz), 6.13 (1H, dd, *J* = 10.3 Hz, *J* = 4.9 Hz), 5.94–5.91 (2H, m), 5.23 (1H, dd, *J* = 7.1 Hz, *J* = 3.4 Hz), 4.41 (1H, d, *J* = 17.1 Hz), 3.87–3.84 (1H, m), 3.78 (1H, d, *J* = 17.1 Hz), 3.60–3.54 (1H, m), 3.51–3.43 (2H, m), 3.37 (3H, s), 2.19–2.13 (1H, m), 2.06–1.97 (1H, m). ^13^C-NMR (125 MHz, CDCl_3_) δ: 163.6, 150.6, 146.8, 146.7, 135.5, 133.9, 130.4, 129.8, 127.6, 126.7, 123.7, 106.7, 103.8, 101.0, 81.8, 72.3, 62.9, 61.3, 60.9, 56.6, 49.3, 28.7. ${\left[\alpha \right]}_{D}^{24}$ = +25.6 (*c* = 0.110; CHCl_3_). EI-MS *m*/*z* (%) 450 (87), 284 (78), 270 (35), 252 (54), 224 (100), 222 (36), 181 (33), 15 (58), 120 (53), 104 (43). ESI-HRMS *m*/*z* calcd for C_24_H_22_N_2_O_7_ [M + H]^+^ 451.1505 found 451.1510.

### 3.3. Biological Assays

#### 3.3.1. *h*AChE and *h*BuChE Inhibition Assay

The *h*AChE and *h*BuChE activities were determined using a modified method of Ellman with acetylthiocholine iodide (ATChI) and butyrylthiocholine iodide (BuTChI) as substrates, respectively [47].

#### 3.3.2. AChE/BuChE Inhibition Mechanism

The procedure for determination of the inhibition mechanism was similar to that for determination of IC_50_, with a difference in that uninhibited and inhibited processes were observed using three different concentrations of ATChI and BuTChI (20 µM, 40 µM, 60 µM). The dependence of absorbance (412 nm) vs. time was measured and the reaction rate was calculated for all reactions (uninhibited and inhibited). Then, a Lineweaver-Burk plot was constructed and Km and Vm values were [38].

#### 3.3.3. GSK-3β Assay

The GSK-3β assay was performed by using Kinase-Glo Kit obtained from Promega (Promega Biotech Iberica, SL, Madrid, Spain) as described previously [39].

#### 3.3.4. CNS Penetration: In Vitro Parallel Artificial Membrane Permeability Assay

The PAMPA assay was performed as previously described [48].

#### 3.3.5. In Vitro Cytotoxicity Study

##### Cell Culture and Culture Conditions

Selected human tumor and non-tumor cell lines {Jurkat (acute T cell leukemia), MOLT-4 (acute lymphoblastic leukemia), A549 (lung carcinoma), HT-29 (colorectal adenocarcinoma), PANC-1 (pancreas epithelioid carcinoma), A2780 (ovarian carcinoma), HeLa (cervix adenocarcinoma), MCF-7 (breast adenocarcinoma), SAOS-2 (osteosarcoma) and MRC-5 (normal lung fibroblasts)} were purchased from either ATCC (Manassas, VA, USA) or Sigma-Aldrich (St. Louis, MO, USA) and cultured according to the provider’s culture method guidelines. All cell lines were maintained at 37 °C in a humidified 5% carbon dioxide and 95% air incubator. Cells in the maximum range of either 10 passages for primary cell line (MRC-5), or in the maximum range of 20 passages for cancer cell lines (Jurkat, MOLT-4, A549, HT-29, PANC-1, A2780, HeLa, MCF-7 and SAOS-2) and in an exponential growth phase were used for this study.

##### WST-1 Cytotoxicity Assay

The WST-1 (Roche, Mannheim, Germany) reagent was used to determine the cytostatic effect of the test compounds. WST-1 is designed for the spectrophotometric quantification of cell proliferation, growth, viability and chemosensitivity in cell populations using a 96-well-plate format (Sigma, St. Louis, MO, USA). The principle of WST-1 is based on photometric detection of the reduction of tetrazolium salt to a colored formazan product. The cells were seeded at a previously established optimal density (30,000 Jurkat, 25,000 MOLT-4, 500 A549, 1500 HT-29, 2000 PANC-1, 5000 A2780, 500 HeLa, 1500 MCF-7, 2000 SAOS-2 and 2000 MRC-5 cells/well) in 100 µL of culture medium, and adherent cells were allowed to reattach overnight. Thereafter, the cells were treated with 100 µL of either corresponding alkaloids or doxorubicin stock solutions to obtain the desired concentrations and incubated in 5% CO_2_ at 37 °C. WST-1 reagent diluted 4-fold with PBS (50 µL) was added 48 h after treatment. Absorbance was measured after 3 h incubation with WST-1 at 440 nm. The measurements were performed in a Tecan Infinite M200 spectrometer (Tecan Group, Männedorf, Switzerland). All experiments were performed at least three times with triplicate measurements at each drug concentration per experiment. The viability was quantified as described previously according to the following formula: (%) viability = (*A*_sample_ − *A*_blank_)/(*A*_control_ − *A*_blank_) × 100, where *A* is the absorbance of the employed WST-1 formazan measured at 440 nm [49]. The viability of the treated cells was normalized to the viability of cells treated with 0.1% DMSO (Sigma-Aldrich) as a vehicle control.

##### Statistical Analysis

The descriptive statistics of the results were calculated and the charts made in either Microsoft Office Excel 2010 (Microsoft, Redmond, WA, USA) or GraphPad Prism 5 biostatistics (GraphPad Software, La Jolla, CA, USA). In this study, all of the values were expressed as arithmetic means with SD of triplicates (n = 3), unless otherwise noted. The significant differences between the groups were analyzed using the Student’s *t*-test and a *P* value ≤ 0.05 was considered statistically significant.

#### 3.3.6. Molecular Modeling Studies

Two structures of *h*AChE and *h*BuChE were gained from RCSB Protein Data Bank–PDB ID: 4EY7 (crystal structure of *h*AChE) and 4BDS (crystal structure of *h*BuChE) [44,45]. All receptor structures were prepared by DockPrep function of UCSF Chimera (version 1.4) and converted to pdbqt-files by AutodockTools (v. 1.5.6) [50,51]. Flexible residues selection was based on previous experience with either *h*AChE or the spherical region around the binding cavity [52,53,54]. Three-dimensional structures of ligands were built by Open Babel (v. 2.3.1), minimized by Avogadro (v 1.1.0) and converted to pdbqt-file format by AutodockTools [55]. The docking calculations were made by Autodock Vina (v. 1.1.2) with the exhaustiveness of 8 [56]. Calculation was repeated 20 times for each ligand and receptor and the best-scored result was selected for manual inspection. The visualization of enzyme-ligand interactions was prepared using The PyMOL Molecular Graphics System, Version 2.0 (Schrödinger LLC, Mannheim, Germany). 2D diagrams were created with BIOVIA, Discovery Studio Visualizer, v 17.2.0.16349 (2016) (Dassault Systèmes, San Diego, CA, USA). 

## 4. Conclusions

In conclusion, twelve derivatives were prepared derived from the β-crinane-type alkaloid haemanthamine. All semisynthetic derivatives were studied for their inhibitory potential of enzymes connected with AD, and for cytotoxicity. Using in silico methods, we were able to identify two plausible binding modes for **1j** and **1m** in *h*AChE and *h*BuChE to provide structural insights into the SAR for the two highlighted compounds reported herein. The potential to penetrate through the BBB of the active derivatives was also studied. The observations made herein should pave the way for the structure-based optimization of novel haemanthamine analogues potentially applicable in neurodegenerative processes. Promising biological activities were demonstrated by **1j**, **1m** and **1g**, and these compounds will be used as lead-structures in the development and optimization of more potent drugs for the treatment of AD in the near future.

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
