# Peer review of "Derivatives of the β-Crinane Amaryllidaceae Alkaloid Haemanthamine as Multi-Target Directed Ligands for Alzheimer’s Disease"

_molecules, 2019, doi:10.3390/molecules24071307_

Round 1
Reviewer 1 Report
"Derivatives of the β-crinane Amaryllidaceae Alkaloid Haemanthamine as Lead Compounds for the Treatment of Alzheimer´s Disease" is well written manuscript for chemical properties but there is a huge incompleteness to accept this in a biological journal.
It is very important to show the effectiveness of the compound/drug in vitro and in vivo system if the title contain a disease name (Alzheimer's disease) and authors are claiming the drug as a therapeutically important.
Also in addition, biological assay part is not well described.what cytotoxity assay did they use? where are the results with explanation?
Author Response
Dears,
we carefully revised our manuscript “Derivatives of the β-crinane Amaryllidaceae Alkaloid Haemanthamine as Lead Compounds for the Treatment of Alzheimer´s Disease” – 474346, according to the reviewers’ suggestions . All changes are highlighted in red.
List of changes and corrections:
Reviewer #1 comment 1+2: Thank you for your kind suggestion. According reviewer´s suggestion, and inspirations from last papers about AD (e.g. Sci Rep 2018, 8:4424) we modified the title of the manuscript. New title sounds: “Derivatives of the β-crinane Amaryllidaceae Alkaloid Haemanthamine as Multi-Target Directed Ligands for Alzheimer´s Disease”.
At the moment, we are unable to run in vivo experiments mainly because of inaccessibility to animals. This will also require some experienced staff and developing appropriate methodology. Even that we believe that these results, as reported herein, will be of interest for Molecules readers.
Reviewer #1 comment 3: The cytotoxicity assay is described in detail in our previous report under Reference [49] (Šafratová, M.; Hošťálková, A.; Hulcová, D.; Breiterová, K.; Hrabcová, V.; Machado, M.; Fontinha, D.; Prudêncio, M.; Kuneš, J.; Chlebek, J.; et al. Alkaloids from Narcissus poeticus cv. Pink Parasol of various structural types and their biological activity. Arch. Pharm. Res. 2018, 41, 208–218. We followed the same methodology in the present study. The results of cytotoxicity assay expressed as cell proliferation are available in Supplementary Material.
Reviewer #3 comments and recommendations
Line 29: change to Twelve derivatives of the β-crinane-type alkaloid haemanthamine (1a-1m) were developed – corrected
46-49: Please define better AD. About the estimated numbers is it AD and other related dementia or just AD? – the definition of AD has been slightly modified.
142 IS compound 1f aromatic? – corrected
190, 192, 195 LogBBB lacks a B – logBB is correct (please see J. Compt. Aided Mo. Des 2011, 25, 1095-1106; Biomed. Appl. Tech. J. 2013,1, 16-34 etc.), we corrected logBBB to logBB in Table 1.
214: it should be table 2 – corrected
221: should be table 2 – corrected
Figure 3B and 4D change HID to HIS
Answer: HID is one of the possible protonation of histidine residue corresponding to its neutral form. This correlates well with the docking methodology perfomed under physiological conditions, i.e. pH = 7.4.
What do the colors of the aminoacids mean in figures 2BD and 4BD?
Anawer: Different colours in Figures 3BD and 4BD illustrate the nature of interactions between amino acid residues and ligands. For instance, orange colour goes for hydrogen bond, green reflects non-specific hydrophobic interaction (e.g. van der Waals forces) and purple/pink means cation-π or π-π interactions. – Headlines of Fig 2 and 4 were slightly modified.
Why is the catalytic triad not depicted in 4B or other figures?
Answer: Catalytic triad residues in 4B and other figures (meaning 3B, 3D and 4D) are displayed only when providing any interactions with the ligands. For the sake of clarity and illustartiive puproses, catalytic triad residues were not omitted in Figures 3A, 3C, 4A and 4C.
Is haemanthamine isolated to perform the reactions? If so please state in the methods.
Information about isolation of haemanthamine for preparation of derivatives was added.
In Scheme 1please check if 1e R group should be CO(CH2)4CH3
The R group of 1e has been corrected.
In Scheme 1 the drawing of aromatic R please add a line to indicate were the aromatic ring is a substituent. (the CH3 joint to the CO is not part of the substituent)
Scheme 1 has been slightly modified.
Reviewer 2 Report
Comments:
The authors present a very interesting paper seeking to develop inhibitors of acetylcholinesterase and butyrylcholinesterase. The background and rationale presented in the paper is well described and very informative. They prepared derivatives of β-crinane-type alkaloid haemanthamine. The synthetic chemistry is well described and would be readily followed. In total 12 new compounds were synthesised and characterised by 1H NMR, 13C NMR and HRMS.
They then assessed their acetylcholinesterase, butyrylcholinesterase and glycogen synthase kinase 3β inhibition activities. They also elucidated the potential of 1g, 1j and 1m to penetrate through the blood-brain barrier. Two potent inhibitors (1j and 1m) were identified and discussion of these data is accurate and appropriate.
They performed molecular docking studies to investigate the binding mode of the most active inhibitors of hAChE and hBuChE.
The data presented support the conclusion drawn, which is useful for others in the related research field, and will be appreciated by the wide readership of Molecules.
Author Response

(The authors gave the same response as above.)

Reviewer 3 Report
Authors synthesize some derivatives of haemanthamine and study enzyme inhibition of AchE, BuchE and GSK3. They also perform in silico binding to explain the results obtained. Finally the PAMPA method shows that the most potent compounds are predicted to cross the BBB. Manuscript is well written with some exceptions marked below. The SAR are interesting and the new activities encountered are as well.
Line 29: change to Twelve derivatives of the β-crinane-type alkaloid haemanthamine (1a-29 1m) were developed.
46-49: Please define better AD. About the estimated numbers is it AD and other related dementia or just AD?
142 IS compound 1f aromatic?
190, 192, 195 LogBBB lacks a B
208: delete have
214: it should be table 2
221: should be table 2
Figure 3B and 4D change HID to HIS
What do the colors of the aminoacids mean in figures 2BD and 4BD?
Why is the catalytic triad not depicted in 4B or other figures?
Is haemanthamine isolated to perform the reactions? If so please state in the methods
In Scheme 1please check if 1e R group should be CO(CH2)4CH3
In Scheme 1 the drawing of aromatic R please add a line to indicate were the aromatic ring is a substituent. (the CH3 joint to the CO is not part of the substituent)
Author Response

(The authors gave the same response as above.)

Round 2
Reviewer 1 Report
Thanks for the revised version of this manuscript.
The cytotoxicity assay needs to be describe here briefly (even though it has been described in previous report).
Author Response
Dears,
we repetitively revised our manuscript “Derivatives of the β-crinane Amaryllidaceae Alkaloid Haemanthamine as Multi-Target Directed Ligands for Alzheimer´s Disease” – 474346, according to the reviewers´ suggestions . All changes are highlighted in green.
List of changes and corrections:
Reviewer #1 comment 1: The cytotoxicity assay needs to be describe here briefly (even though it has been described in previous report).
Answer: Shorten description of cytotoxicity assay has been incorporated into revised manuscript.
Best wishes
Lucie Cahlíková, Dr.